# Low dose of luteolin activates Nrf2 in the liver of mice at start of the active phase but not that of the inactive phase

Tomoya Kitakaze, Atsushi Makiyama, Yoko Yamashita, Hitoshi Ashida *

Department of Agrobioscience, Graduate School of Agricultural Science, Kobe University, Kobe, Hyogo, Japan

* ashida@kobe-u.ac.jp

**Data Availability Statement:** All relevant data are within the paper and its Supporting Information files.

## Abstract

A flavone luteolin has various health-promoting activities. Several studies reported that high dose of luteolin activates the Nrf2/ARE pathway in the liver. However, the effect of the low dose of luteolin that can be taken from a dietary meal on the Nrf2 activation remain unclear. It is expected that the flavonoid metabolism possesses a circadian rhythm, since nutritional metabolism processes daily cycle. In this study we investigated whether an administration affects the Nrf2 activation. ICR mice were orally administered 0.01–10 mg/kg body weight of luteolin once a day for 7 days at two time-points: at the start of active phase (ZT12) or at that of inactive phase (ZT0). Luteolin increased the nuclear translocation of Nrf2, resulting in the increases in its target gene products HO-1 and NQO1 at ZT12 but not at ZT0. The expression level of Nrf2 was lower at ZT12 than at ZT0 in the liver. We also found that the level of luteolin aglycon in the plasma is higher at ZT12 than at ZT0. These results suggest that the low dose of luteolin can activate Nrf2 pathway and the aglycon form of luteolin may mainly contribute to activate the Nrf2 pathway at ZT12 in the liver.

## Introduction

Circadian rhythms are daily cycle of internal processes that regulates behavior, physiology, and metabolism and repeats roughly every 24 hours. Nutritional metabolism processes in both gastrointestinal tract and peripheral tissue also have a circadian rhythm. Circadian rhythmicity of nutritional metabolism processes is regulated by the core molecular clock, which consists of a transcriptional-translational feedback loop. The transcriptional activators brain and muscle Arnt-like protein-1 (BMAL1) and circadian locomotor output cycles kaput (CLOCK) induce expression of their own repressors cryptochrome (CRY) and period (PER), resulting in generating 24 hours oscillations. Several human studies revealed correlation between irregular eating times and the risk of obesity and insulin resistance [1,2], and restriction of feeding time contributes to prevent and ameliorate these diseases [3,4]. Thus, feeding time is important to maintaining for human health. On the other hand, many studies using culture cells and animals reveal that plants-derived phenolic compounds, in particularly flavonoids show many

**Funding:** This work was supported by JSPS KAKENHI Grant Number 17H00818 (to H. A.). https://www.jsps.go.jp/english/e-grants/index.html The funders had no role in study design, data collection and analysis, decision to publish, or preparation of the manuscript.

**Competing interests:** The authors have declared that no competing interests exist.

beneficial functions for human health. However, most of study have not considered feeding time of flavonoids. Thus, the involvement of feeding time on the function of flavonoids remain unclear.

Flavonoids possess various health-promoting activities as antioxidants. Evidences from the cultured cells and animal studies revealed that individual flavonoids have many health benefits, such as prevention of oxidative stress, inflammation and lipid accumulation [5,6]. Epidemiological studies suggest that high dietary intake of flavonoids is associated with lower risk of diseases including cardiovascular disease and several types of cancer [7,8]. The mean daily intake of flavonoids has been estimated in many countries; e.g., US (207.0 mg/day), Korea (318.0 mg/day) and Spain (376.69 mg/day) [9–11]. A number of animal studies are intended to evaluate the effects of high dose of flavonoids (more than 10 mg/kg), but little is known about the function of low dose of flavonoids that can be taken from a dietary meal.

Many flavonoids have antioxidant activity not only by its free radical scavenging ability but also by inducing the expression of antioxidant enzymes. Nuclear factor-erythroid-2-related factor 2 (Nrf2) belongs to the cap'n'collar basic leucine zipper family, and is a key regulator of oxidative stress in numerous types of cells, as well as hepatocytes. Nrf2 is primarily regulated by Kelch-like ECH-associated protein 1 (Keap1), a substrate adaptor for a cul3-containing E3 ubiquitin ligase [12]. Under the normal condition, Nrf2 interacts with Keap1 in the cytoplasm and is rapidly degraded by the ubiquitin-proteasome pathway [13]. However, under oxidative stress, Nrf2 leads to its dissociation from Keap1, subsequent translocates into the nucleus, binds to DNA motif known as an antioxidant response element (ARE) sequences, and enhances the transcription of ARE-responsive genes such as heme oxygenase-1 (HO-1) and NAD(P)H:quinone oxidoreductase 1 (NQO1) [14,15]. In recent years, many studies have suggested that an Nrf2-dependent antioxidant response is necessary for the cells to protect against some insults from the environment. Many studies using Nrf2 knockout or constitutive Nrf2 activating mice demonstrated that Nrf2 regulates cellular detoxification, mitochondrial respiratory chain activity, regeneration capacity, and developing several diseases [16–18]. Therefore, Nrf2 have emerged as a promising strategy for prevention and amelioration of oxidative stress related diseases.

Luteolin (3', 4', 5, 7-tetrahydroxyflavone) is a plant derived flavone and exists in human circulating blood after intake of edible plants. A daily ordinary meal in Japan and China contains approximately 0.2 mg and 8.08 mg of luteolin, respectively [19,20]. The serum concentration of luteolin reaches about 100 nM by dietary habit [19]. Luteolin possesses a lot of physiological functions including antioxidative activity [21,22], while the number of cultured cells and animal studies evaluate the effects in the concentration that exceeds the amount taken from the daily meal. On the other hand, our previous study revealed that the physiological nanomolar-concentration range of luteolin induces Nrf2/ARE pathway in human hepatoma HepG2 cells [23]. However, the effect of the low dose of luteolin that can be taken from a dietary meal on the Nrf2 activation remain unclear.

After intake of flavonoids, including luteolin, they are metabolized to glucuronide, sulfate and methylated conjugates by glucuronosyltransferases (UGTs), sulfotransferases (SULTs), and catechol O-methyl transferases (COMTs), respectively [24–27]. Since the expression of the part of these enzymes exhibits circadian rhythm [28,29], it is expected that the amounts and components of flavonoid metabolites are changed depending on the circadian rhythm. In this study, we investigated whether the low dose of luteolin (0.01 to 10 mg/kg) activate Nrf2 pathway and involvement of its administration timing on the induction of antioxidant enzymes in the liver of mice.

## Materials and methods

### Materials

Luteolin was obtained from Wako Pure Chemical (Osaka, Japan). Antibodies against NQO1, CLOCK and PER2 were from Abcam (Cambridge, UK). Antibodies against β-actin and BMAL1 were from Cell Signaling Technology (Danvers, MA, USA). Antibodies against HO-1 (Enzo Life Sciences Inc., Farmingdale, NY, USA), lamin B (Santa Cruz Biotechnology, Dallas, TX, USA) and Nrf2 (Medical & Biological Laboratories, Aichi, Japan) were also used in this study. All other reagents used were of the highest grade available from a commercial source.

### Animal experiments

All animal experiments were approved by the Institutional Animal Care and Use Committee (Permission # 27-05-09) and carried out according to the guidelines for animal experiments at Kobe University. Seventy male, 6-week-old ICR mice (Japan SLC, Inc., Shizuoka, Japan) were obtained from Japan SLC, Inc. (Shizuoka, Japan) and allowed free access to tap water and a purified diet AIN-93M (Research Diets, NJ, USA) with a 12:12–h light/dark cycle (light period starting from 8:00 AM; equal to Zeitgeber time (ZT) 0) at a controlled temperature ($25 \pm 3°C$). To examine the effect of luteolin on the expression of phase II drug-metabolizing enzymes, mice were randomly assigned to two groups. One group was administrated luteolin at ZT22 and another group was administrated it at ZT10. Each groups of mice were also randomly divided into 5 groups of 6–8 each and were orally administrated different concentration of luteolin 0.01, 0.1, 1 or 10 mg/kg body weight (B.W.) and propylene glycol as a vehicle for 7 days. The mice were killed by exsanguination following cardiac puncture 2 h after the final luteolin administration. The plasma and livers were collected. Tissues and plasma were frozen in liquid $N_2$ and stored at -80°C until analyzed.

### Preparation of tissue lysates and nuclear fractions

The liver of the mice was homogenized with RIPA buffer consisting of 50 mM Tris-HCl; pH 8.0, 150 mM NaCl, 1% (v/v) NP-40, 0.5% (w/v) deoxycholic acid, 0.1% (w/v) sodium dodecyl sulfate (SDS), and 0.5 mM dithiothreitol (DTT), and protease inhibitors [(1 mM phenyl-methylsulfonyl fluoride (PMSF), 5 μg/mL leupeptin, and 5 μg/mL aprotinin)] and phosphatase inhibitors (10 mM NaF and 1 mM $Na_3VO_4$), and left on ice for 1 h with occasional mixing. The homogenates were centrifuged at $20,000 \times g$ for 20 min at 4°C and obtained supernatants were used as tissue lysates.

To prepare the unclear fractions, the liver of the mice was homogenized with lysis buffer consisting of 20 mM HEPES; pH 7.6, 20% glycerol, 10 mM NaCl, 1.5 mM $MgCl_2$, 0.2 mM EDTA, 0.5 mM DTT and the same protease- and phosphatase-inhibitor cocktail. The mixtures were centrifuged at $800 \times g$ for 10 min at 4°C, and the precipitate was suspended in hypertonic buffer consisting of 20 mM HEPES; pH 7.6, 20% glycerol, 420 mM NaCl, 1.5 mM $MgCl_2$, 0.2 mM EDTA, 0.5 mM DTT, and the same protease- and phosphatase-inhibitor cocktail. After left on ice for 1 h, the mixture was further centrifuged at $20,000 \times g$ for 20 min at 4°C, and the resultant supernatants were used as the nuclear fractions.

### Western blotting analysis

The nuclear fraction was used for the detection of Nrf2 and lamin B. Other factors were detected in tissue lysate. After electrophoresis with SDS polyacrylamide gel, the proteins in the gel were transferred onto a polyvinylidene fluoride (PVDF) membrane (GE Healthcare Bio-Science Cooperation, New Jersey, USA). The membrane was treated with a commercial

blocking solution (Blocking One, Nacalai Tesque, Kyoto, Japan) for 1 h at room temperature. The membrane was incubated with primary antibodies overnight at 4˚C, followed by incubation with the corresponding HRP-conjugated secondary antibody for 1 h at room temperature. The blots were developed using Immuno Star LD Western Blotting Substrate (Wako Pure Chemical Industries) and immunocomplexes were detected with Light-Capture II (ATTO Co., Tokyo, Japan). The density of the specific band was determined using ImageJ image analysis software (National Institutes of Health, Bethesda, MD, USA).

## RNA isolation and quantitative real-time PCR

The total RNA from the liver of mice were isolated using TRIzol reagent (Invitrogen, CA, USA) in accordance with the manufacturer's instructions, and subjected to the reverse transcriptional reaction. The resultant cDNA was subjected to quantitative real-time PCR using the SYBR Pre-mixEx Taq II (Takara Bio., Kyoto, Japan) and a two-step PCR method on a real-time PCR system (TAKARA PCR Thermal Cycler Dice, Takara Bio, Shiga, Japan). The following specific primers were used: *Gapdh* (forward primer 5´-CATGGCCTTCCGTGTTCCTA-3´ and reverse primer 5´-CCTGCTTCACCACCTTCTTGA-3´); *Arntl* (forward primer 5´-TCAGATGACG AACTGAAACACC-3´ and reverse primer 5´-CGGTCACATCCTACGACAAA-3´); *Clock* (forward primer 5´-CCAGTCAGTTGGTCCATCATT-3´ and reverse primer 5´-TGGCTCCTAA CTGAGCTGAAA-3´); *Per2* (forward primer 5´-CAACACAGACGACAGCATCA-3´ and reverse primer 5´-TCCTGGTCCTCCTTCAACAC-3´); *Nfe2l2* (forward primer 5´-CTCCTTG AGCTCAAATCCCACCTTA-3´ and reverse primer 5´-TGGGCTCTGCTATGAAAGCA-3´); *Hmox1* (forward primer 5´-CCTCACTGGCAGGAAATCATC-3´ and reverse primer 5´-CCT CGTGGAGACGCTTTACATA-3´); and *Nqo1* (forward primer 5´-TTCTCTGGCCCGATTCAG AGT-3´ and reverse primer 5´-GGCTGCTTGGAGCAAAATAG-3´). *Gapdh* mRNA was used as a normalized control. The relative gene expression level was calculated by the comparative cycle threshold method.

## Quantitative analysis of luteolin in the plasma of mice

The amount of luteolin in the plasma of mice was quantified by a high performance liquid chromatography (HPLC). The plasma was treated with or without deconjugation enzymes glucuronidase and sulfatase. Briefly, an aliquot of 150 μl of plasma was mixed with 15 μl of 20% ascorbic acid and 75 μl of 75 mM phosphate buffer; pH 6.8 containing with or without 500 U β-glucuronidase, and incubated for 1 h at 37˚C. Then, the mixture was mixed with 75 μl of 200 mM sodium acetate buffer; pH 5.0 containing with or without 10 U sulfatase, and further incubated for 1 h at 37˚C. To extract the luteolin, ethyl acetate was added to the reaction mixture and the mixture was vigorously mixed for 30 s. After centrifugation at $1000 \times g$ for 10 min, the ethyl acetate layer was collected, evaporated with a centrifugal concentrator, and re-dissolved in 50% methanol. The HPLC system was consisting of a Shimadzu liquid chromatograph model CBM-20A (Kyoto, Japan) equipped with an autosampler using a Cadenza CL-C18 column ($250 \times 4.6$ mm inner diameter, 3 μm particle diameter; Imtakt, Kyoto, Japan) at a flow rate of 0.7 ml/min, column temperature of 35˚C, and UV detection at 350 nm. The mobile phase was consisting of solvents A (0.1% formic acid in $H_2O$) and B (100% acetonitrile). The gradient program was as follows: the initial composition consisted of 70% A and 30% B; followed by a linear gradient to 80% B over 40 min.

## Statistical analysis

All data are expressed as means ± standard error (SE). One-way analysis of variance (ANOVA) with the Dunnett's post hoc test was used in the experiments that hat three or more

groups to determine the significant differences between the treatment and control groups. The significant difference between the two groups was determined using the Student's *t* test. Statistical analysis was performed with JMP statistical software version 11.2.0 (SAS Institute, Cary, NC, USA). The level of statistical significance was set as $p < 0.05$.

## Results

### Difference of the effect of luteolin administration at between ZT0 and ZT12 in the liver

The effect of luteolin on the expression of antioxidant enzymes, HO-1 and NQO1 at the different time-points was firstly assessed. As shown in Fig 1, administration of luteolin at 0.1, 1 and 10 mg/kg induced the expression of HO-1 and NQO1 at ZT12 ($p < 0.05$). In contrast, luteolin had no effect on the expression of these enzymes at ZT0 ($p > 0.05$). These results indicate that the effect of luteolin administration on the expression of the antioxidant enzymes is different at ZT0 and ZT12.

### Expression of clock genes and antioxidant enzymes in the liver at ZT0 and ZT12

To evaluate the expression of Nrf2 and its target gene, mRNA expression and protein level of Nrf2 and HO-1 were determined in the liver of control mice at ZT0 and ZT12. Both mRNA and protein expression of Nrf2 and HO-1 were higher at ZT0 than at ZT12 ($p < 0.05$) (Fig 2A and 2B). The abundance of Nrf2 in the nucleus was also higher at ZT0 than at ZT12 ($p < 0.05$) (Fig 2C). The expression of Keap1 was higher at ZT0 than at ZT12 ($p < 0.05$) (Fig 2D). These results suggest that the transcriptional activity of Nrf2 is higher at ZT0 than ZT12 and Nrf2 activity is regulated by transcription of mRNA. However, the level of Nrf2 ubiquitination and interaction with Keap1 remain unclear. Further studies are needed in the future to clarify this issue. The mRNA expression of *Arntl* and *Clock* were higher in ZT0 than in ZT12 ($p < 0.05$), and that of *Per2* was lower at ZT0 than at ZT12 ($p < 0.05$), suggesting circadian rhythm was normal in these mice (S1 Fig). These results suggest that the mRNA expression of Nrf2 and its target gene, HO-1 have a circadian rhythm in the liver.

### Effect of luteolin on the activation of Nrf2 in the liver

To evaluate the effect of luteolin on the activation of the Nrf2 in the liver, the expression of Nrf2 in nucleus was determined. As shown in Fig 3, administration of luteolin at 0.1, 1 and 10 mg/kg significantly increased nuclear translocation of Nrf2 to 1.65 ± 0.13, 1.80 ± 0.10 and 1.74 ± 0.23 folds, respectively, compared with control (0 mg/kg) at ZT12 ($p < 0.05$). Since nuclear translocation of Nrf2 was the highest at 1 mg/kg luteolin, we next evaluated the effect of 1 mg/kg luteolin on the mRNA expression of Nrf2 and its target proteins. The administration of luteolin at 1 mg/kg increased mRNA expression levels of *Hmox1* and *Nqo1* in the liver at ZT12 to 1.33 ± 0.07 and 1.43 ± 0.13 folds compared with control (0 mg/kg) ($p < 0.05$) (Fig 4). However, luteolin had no effect on mRNA expression levels of *Nfe2l2* ($p > 0.05$) (Fig 4). These results suggest that administration of luteolin activates the Nrf2/ARE pathway and following upregulation of the expression of its target genes, HO-1 and NQO1 through increasing nuclear translocation of Nrf2 but not induction of the expression of Nrf2 mRNA.

### Difference of the plasma luteolin level between at ZT0 and at ZT12

Finally, we quantified the plasma concentration of luteolin and its conjugates after administration of 1 mg/kg luteolin at ZT0 and ZT12 by the high performance liquid chromatography

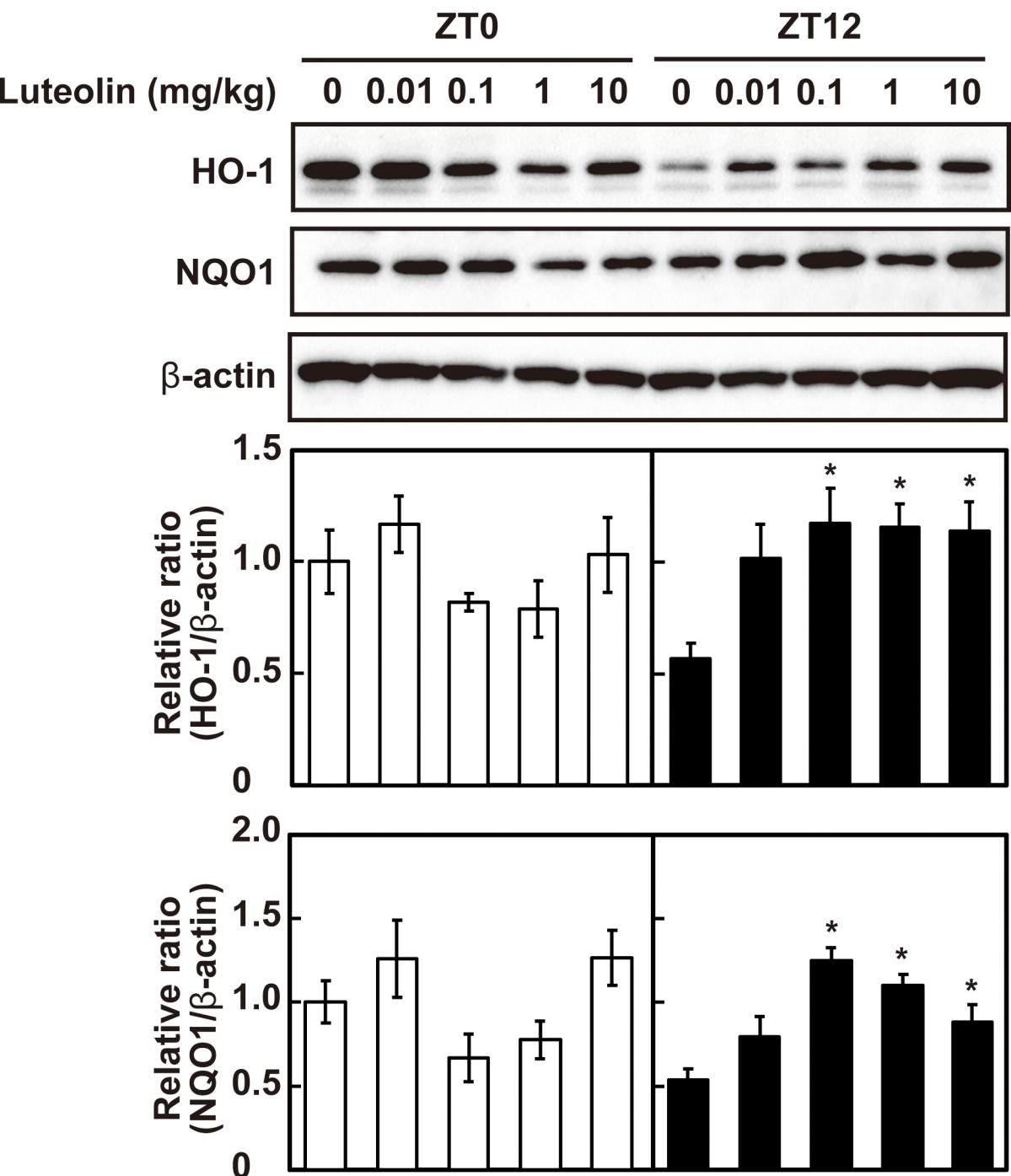

**Fig 1. Effect of luteolin on the expression of antioxidant enzymes at different time points.** The expression levels of HO-1 and NQO1 were determined at ZT0 and at ZT12 by western blotting and normalized by that of β-actin. The intensity of each band was quantified by ImageJ 1.44. The results are represented as the mean ± SE (n = 6–8). Asterisks indicate a significant difference from the control by Dunnett's test ($p < 0.05$).

(HPLC). As shown in Fig 5, the concentration of luteolin aglycon in the plasma at ZT12 was significantly higher than that at ZT0 to 83.04 ± 16.66 nM and 15.79 ± 3.15 nM, respectively ($p < 0.05$). The concentration of luteolin conjugate in the plasma was not significant changed

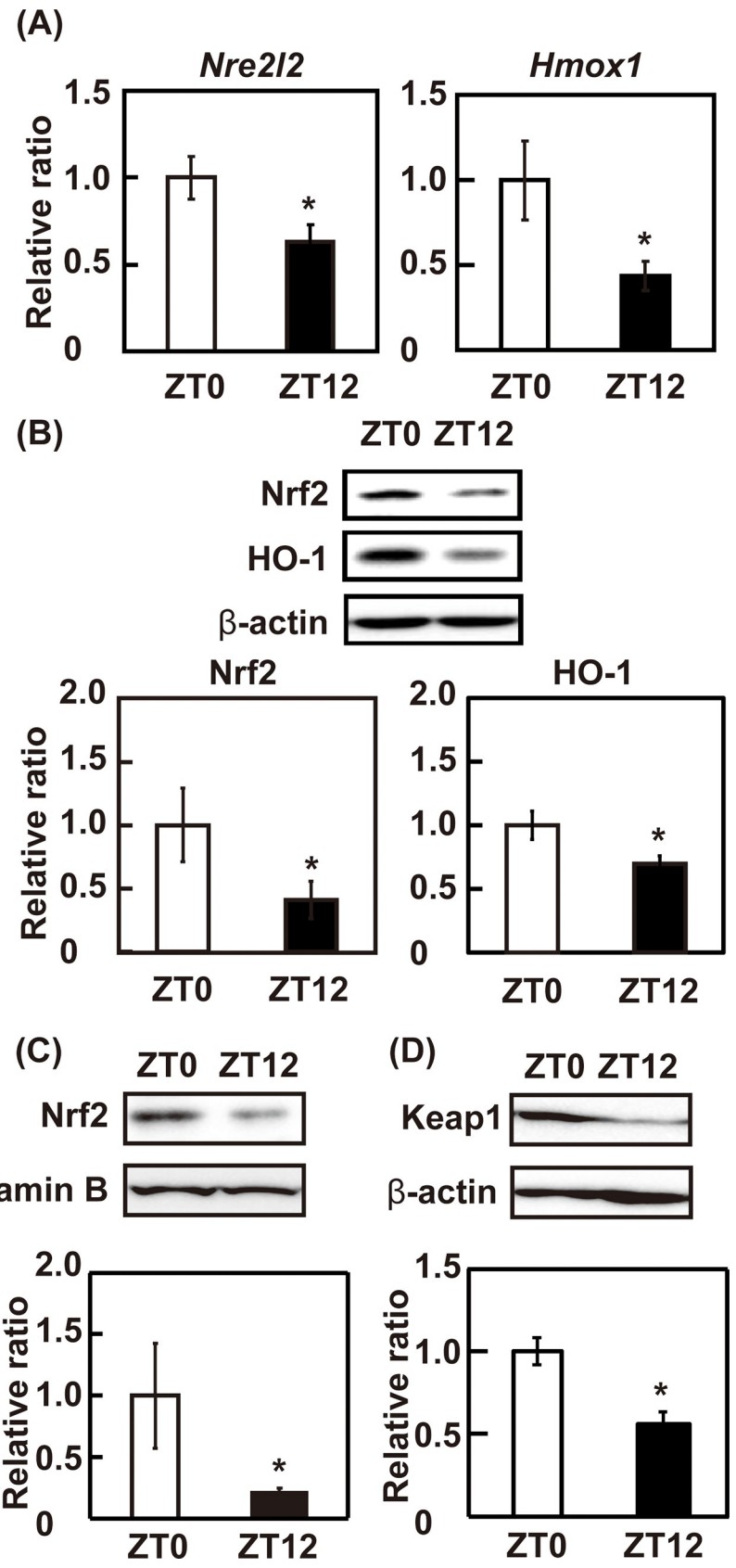

**Fig 2. The expression levels of clock genes and antioxidant enzymes at different time-points.** (A) mRNA and (B) protein levels of Nrf2 and HO-1 were determined by real-time PCR and western blotting, respectively. The mRNA expression level was normalized by the expression of *Gapdh*. The protein expression level was normalized by the expression level of β-actin, and the intensity of each band was quantified by ImageJ 1.44. (C) The expression level of nuclear Nrf2 was determined by western blotting and normalized by that of lamin B. The intensity of each band was quantified by ImageJ 1.44. (D) The expression level of cytosolic Keap1 was determined by western blotting and normalized by that of β-actin. The intensity of each band was quantified by ImageJ 1.44. The results are represented as the mean ± SE (n = 6–8). Asterisks indicate a significant difference from ZT0 by Dunnett's test ($p < 0.05$).

but about 1.7-fold higher at ZT12 compared with at ZT0 to 657.20 ± 147.51 nM and 382.75 ± 63.23 nM, respectively ($p > 0.05$). The total concentration of luteolin (aglycon + conjugates) was also not significant changed but about 1.8-fold higher at ZT12 compared with at ZT0 to 740.23 ± 159.40 nM and 401.26 ± 66.36 nM, respectively ($p > 0.05$). These results suggest that aglycon form of luteolin may mainly contribute to activate Nrf2 pathway at ZT12 in the liver.

## Discussion

In the present study, we demonstrated that the low dose of luteolin induced antioxidant enzymes through the activation of Nrf2 in the liver of mice at ZT12 but not at ZT0. We also found concentration of luteolin aglycon in the plasma at ZT12 was significantly higher than that at ZT0. A daily ordinary meal in Japan contains approximately 0.2 mg/day [20]. Cao et al. showed that the daily intake estimates of luteolin amounted to 8.08 mg/day in Harbin, China [19]. Under these conditions, the dose of luteolin for mice was calculated as 0.04–1.7 mg/kg once a daily according to the human equivalent dose calculation [30]. Luteolin at 0.1 and 1 mg/kg increased the expression of HO-1 and NQO1 and nuclear accumulation of Nrf2, and luteolin at 0.01 mg/kg tended to increase nuclear accumulation of Nrf2 (Figs 1 and 3) at ZT12. These results suggested that the low concentration of luteolin that can be taken from a dietary

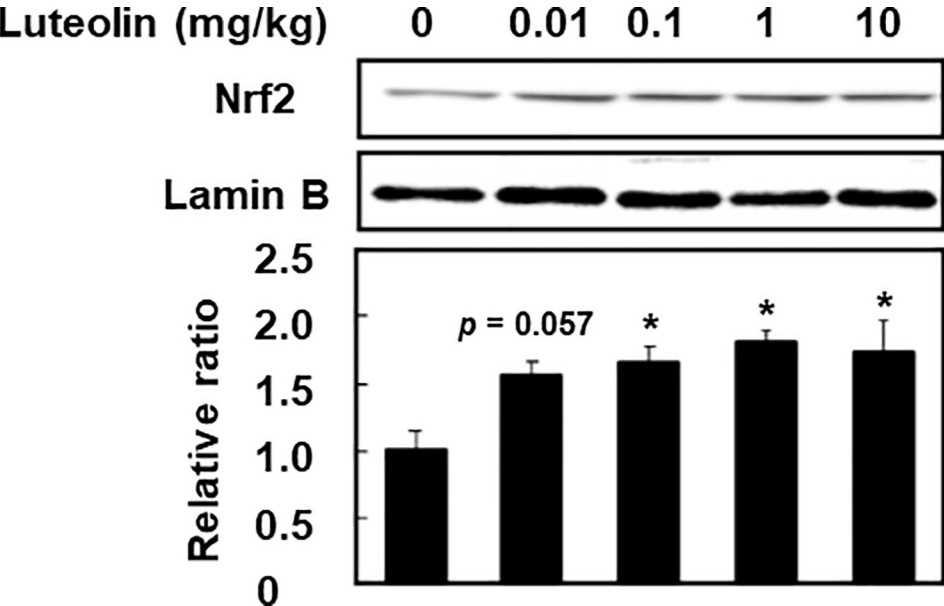

**Fig 3. Effect of luteolin on the nuclear translocation of Nrf2 in the liver at ZT12.** The expression level of nuclear Nrf2 was determined by western blotting and normalized by that of lamin B, and the intensity of each band was quantified by ImageJ 1.44. The results are represented as the mean ± SE (n = 6–8). Asterisks indicate a significant difference from the control by Dunnett's test ($p < 0.05$).

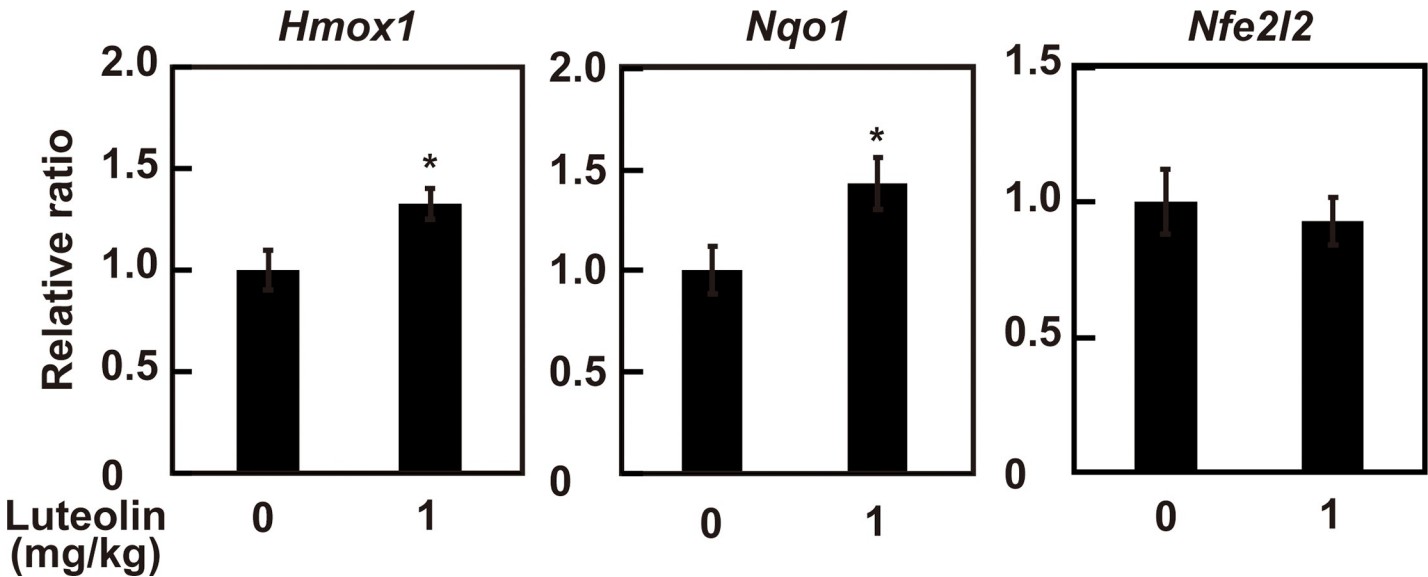

**Fig 4. Effect of luteolin on the mRNA expression of antioxidant genes in the liver at ZT12.** The livers from mice supplemented with 0 and 1 mg/kg luteolin were used. The mRNA levels of Nrf2, HO-1 and NQO1 were determined by real-time PCR. The results are represented as the mean ± SE (n = 6–8). Asterisks indicate a significant difference from the control by the Student's *t* test ($p < 0.05$).

meal can activate Nrf2, resulting in an increase the expression of its target genes HO-1 and NQO1 at the start of active phase. The activation of Nrf2 is regulated by multiple steps, such as transcription, translation and protein stability. Our previous study has demonstrated that luteolin at the physiological concentration range activates Nrf2 nuclear translocation by reduction of Nrf2 ubiquitination through increasing modified Keap1 and phosphorylation of Nrf2 in HepG2 cells [23]. This result supports our finding that the low dose of luteolin increased nuclear accumulation of Nrf2 independently of its mRNA transcription in the liver.

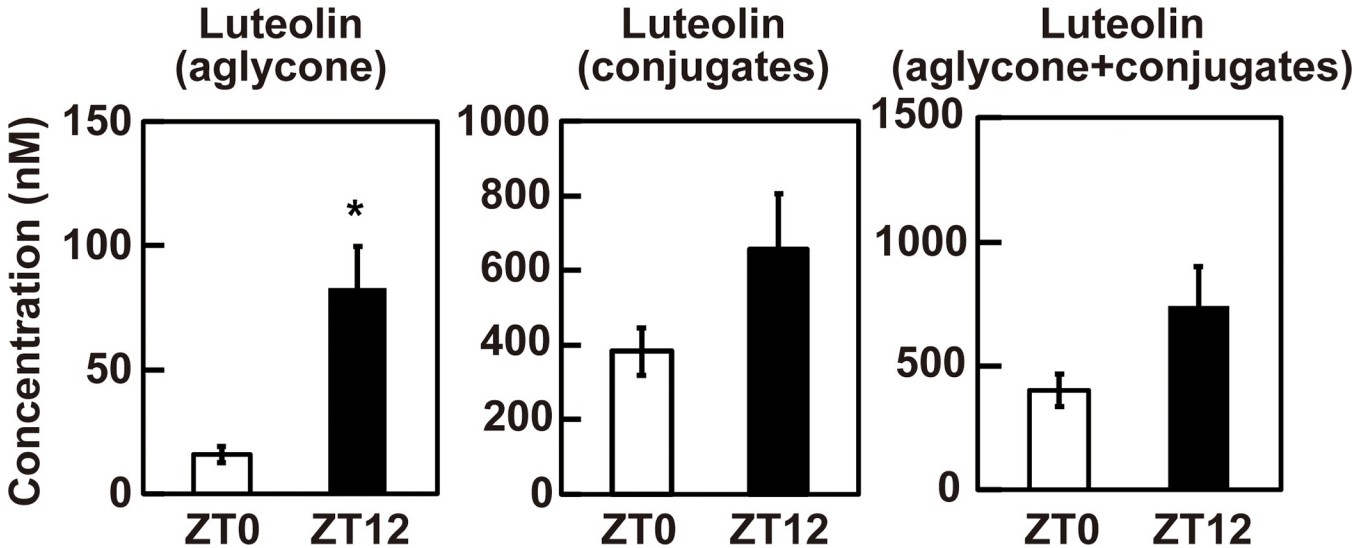

**Fig 5. The plasma concentration of luteolin and its conjugates at ZT0 and at ZT12.** The plasma from mice supplemented with 0 and 1 mg/kg luteolin were used. The plasma concentration of luteolin aglycon, conjugates and aglycon + conjugates were determined by HPLC. The results are represented as the mean ± SE (n = 6–8). Asterisk indicates a significant difference from ZT0 by the Student's t test ($p < 0.05$).

Luteolin glucuronides mainly exist in plasma and organs after oral administration of luteolin in rats, whereas luteolin sulfate, luteolin-3′-O-sulfate, was mainly detected from plasma after intake of luteolin in humans [26]. In mice, luteolin glucuronides are major metabolites existed in the plasma after oral administration of luteolin. Anti-inflammatory effect of luteolin aglycone is higher than its glucuronides, such as luteolin-4′-O-glucuronide, luteolin-3′-O-glucuronide and luteolin-7-O-glucuronide in lipopolysaccharide-treated mouse macrophage-like RAW264.7 cells [31]. Anti-inflammatory effect of luteolin-3′-O-sulfate is lower than that of luteolin aglycone and luteolin glucuronides [26]. Methylated luteolins are generated by COMT in HepG2 cells after treated with luteolin. Luteolin metabolites-derived from HepG2 cells slightly inhibits lipopolysaccharide-induced inflammatory cytokines, and that-derived from co-treated with COMT inhibitor enhances the inhibitory effect against lipopolysaccharide in RAW264.7 cells [32]. Furthermore, inhibitory effect of luteolin aglycone on transthyretin-induced cytotoxicity is higher than luteolin glucuronide in human blastoma SH-SY5Y cells [33]. These results suggest that luteolin aglycone acts as an active form to perform physiological function in the body, and support our finding that luteolin activates Nrf2 at the start of active phase of mice with high plasma level of luteolin aglycon. However, it remains unclear that whether luteolin metabolites fail to induce Nrf2 activation in the liver. Further study is needed in the future.

The absorption and metabolism of proteins, carbohydrates, and lipids exhibit circadian rhythmicity [34]. It is reported that the expression levels of UGTs and SULTs exhibit circadian rhythm in the liver of mice [35]. The expression levels of SULT isoforms, *Sult1a1*, *Sult1d1*, and *Sult5a1*, and UGT isoforms, *Ugt2a3*, *Ugt2b1* and *Ugt2b36*, are higher around the start of active phase, at ZT9 or at ZT13, than that of inactive phase, whereas the expression levels of *Ugt2b34* and *Ugt1a5* are highest at ZT1 and ZT21, respectively [35]. A part of flavonoid glucuronides is deconjugated to its aglucone by β-glucuronidase [36,37]. This process is important to exert a physiological function of flavonoid in the body. Under inflammatory conditions, β-glucuronidase is secreted from macrophages and neutrophils, and metabolites flavonoid-glucuronide to flavonoid aglycone, resulting in exerting physiological function [36,38]. Recently it has been reported that the cell number and functions of macrophages and neutrophils have a cycle approximately 24 hours [39,40]. These results suggest that circadian rhythms involve in the conjugation and deconjugation reactions of the flavonoids, resulting in regulating the biological activity of flavonoids including luteolin.

Our study showed that expression level of Nrf2 was higher at ZT0 than at ZT12 in the liver. In accordance with our study, the expression level of the Nrf2 shows circadian rhythm, and is higher at the start of inactive phase than that of the active phase in lung of mice [41]. On the other hand, both Nrf2 knockout mice and Keap1 knockout mice which is regarded as Nrf2 constitutive active mice, demonstrate disruption of circadian rhythm [42]. A Nrf2 activator, tertiary butylhydroquinone, also affects clock gene expression [42]. These reports suggest that luteolin can regulate the circadian rhythm though the Nrf2 activation. Indeed, it has been reported that 200 μM luteolin decreases the amplitude of clock genes expression in mouse embryonic fibroblasts [43].

Chaix et al. have revealed that time-restricted feeding, food access restricted to 10 h during the dark phase, prevents obesity and metabolic syndrome in mice lacking a circadian clock, suggesting feeding time is critical for maintaining metabolic homeostasis [44]. Resveratrol, a polyphenol found in grapes, possesses anti-oxidant effect against reactive oxygen species and oxidative stress [45], but this effect is dosing-time dependent. Gadacha et al. have reported that resveratrol acts as an antioxidant during the active phase and as a pro-oxidant during the inactive phase of rats [46], suggesting certain polyphenols have different effects depending on the intake timing. Several reports have revealed that high concentration of luteolin (80–100 mg/

kg) attenuates the liver injury and hepatotoxicity through the activation of Nrf2 [21,22]. Taken together, these results suggest that luteolin intake at the start of active phase may possess the powerful effect for prevention and amelioration to several diseases occurring from oxidative stress through the activation of Nrf2 even if the amount of luteolin is very low. Future clinical trials are needed to extend this finding to humans since research subject is limited to nocturnal animal in this study.

## Conclusions

In this study, we demonstrated that different administration timings of luteolin affects the activation of Nrf2 in the liver of mice. We concluded that the higher plasma level of luteolin aglycon at the start of active phase compared to that of inactive phase contributes a luteolin efficiency in mice. Our findings suggest that the intake timing of flavonoids is important to perform more powerful physiological functions.

## Supporting information

**S1 Fig. The expression levels of clock genes at different time-points.** mRNA levels of *Arntl*, *Clock* and *Per2* were determined by real-time PCR. The mRNA expression level was normalized by the expression of *Gapdh*. The results are represented as the mean ± SE (n = 6–8). Asterisks indicate a significant difference from ZT0 by the Student's *t* test ($p < 0.05$).
(TIF)

**S1 Data.**
(PDF)

**S1 Raw Images.**
(PDF)

## Author Contributions

**Conceptualization:** Tomoya Kitakaze, Hitoshi Ashida.

**Data curation:** Tomoya Kitakaze, Atsushi Makiyama.

**Formal analysis:** Tomoya Kitakaze, Yoko Yamashita.

**Funding acquisition:** Hitoshi Ashida.

**Supervision:** Hitoshi Ashida.

**Writing – original draft:** Tomoya Kitakaze.

**Writing – review & editing:** Hitoshi Ashida.

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
