## [Decision Letter · Decision Letter 0]

20 Feb 2020

PONE-D-20-01855

Low dose of luteolin activates Nrf2 in the liver of mice dependent on an administration timing: luteolin activates Nrf2 at the start of the active phase but not that of inactive phase of mice

PLOS ONE

Dear Dr. Ashida,

Thank you for submitting your manuscript to PLOS ONE. After careful consideration, we feel that it has merit but does not fully meet PLOS ONE’s publication criteria as it currently stands. Therefore, we invite you to submit a revised version of the manuscript that addresses the points raised during the review process.

We would appreciate receiving your revised manuscript by Apr 05 2020 11:59PM. To enhance the reproducibility of your results, we recommend that if applicable you deposit your laboratory protocols in protocols.io, where a protocol can be assigned its own identifier (DOI) such that it can be cited independently in the future. For instructions see: http://journals.plos.org/plosone/s/submission-guidelines#loc-laboratory-protocols

We look forward to receiving your revised manuscript.

Kind regards,

Nobuyuki Takahashi, Ph.D.

Academic Editor

PLOS ONE

Journal Requirements:

3. As part of your revision, please complete and submit a copy of the ARRIVE Guidelines checklist, a document that aims to improve experimental reporting and reproducibility of animal studies for purposes of post-publication data analysis and reproducibility: https://www.nc3rs.org.uk/arrive-guidelines. Please include your completed checklist as a Supporting Information file. Note that if your paper is accepted for publication, this checklist will be published as part of your article.

4. To comply with PLOS ONE submissions requirements, in your Methods section, please provide additional information on the animal research and ensure you have included details on (1) methods of anesthesia and/or analgesia, and (2) efforts to alleviate suffering.

Reviewers' comments:

Reviewer's Responses to Questions

**Comments to the Author**

1. Is the manuscript technically sound, and do the data support the conclusions?

Reviewer #1: Yes

Reviewer #2: Yes

2. Has the statistical analysis been performed appropriately and rigorously? 

Reviewer #1: No

Reviewer #2: Yes

3. Have the authors made all data underlying the findings in their manuscript fully available?

Reviewer #1: Yes

Reviewer #2: Yes

4. Is the manuscript presented in an intelligible fashion and written in standard English?

Reviewer #1: Yes

Reviewer #2: Yes

5. Review Comments to the Author

Reviewer #1: This manuscript is very interesting for relationship of the luteolin-induced Nrf activation and circadian rhythms in vivo. In fact, author showed that low-dose luteolin activates Nrf2 and increase plasma luteolin levels in active phase. However, I have a few questions. Thus, I propose that to improve the manuscript and to increase potential impact figures in this field of scientific research.

Major comment

1: In Fig.1, author showed that protein levels of Nrf2 and HO-1 were induced in ZT12, but not ZT0. However, this can’t be investigated. Because Nrf2 levels in control level (luteolin 0 mg/kg) is different at ZT0 and ZT12. If author wish to compare the two groups, author should be measure the protein expressions on the same membrane.

2: In Fig.2, author indicated mRNA level of Nrf2 was regulated by circadian rhymes. This point is very interested. However, I have a question. Why did Nrf2 protein level decrease in ZT12 compared to ZT0? Author should be measure Nrf2 levels in nuclear fraction and keap1 protein level in cytoplasm.

3: For statistical analysis, it is better to do a statistical test using one-way analysis but not Dunnett’s test. Because various concentration of luteolin treated for mice in this animal experiment. This point is very important.

Reviewer #2: The authors of 'Low dose of luteolin activates Nrf2 in the liver of mice dependent on an administration timing: luteolin activates Nrf2 at the start of the active phase but not that of inactive phase of mice' present a structured manuscript, focusing on the effects of low dose of luteolin in mice liver, especially with regard to its regulation of Nrf2 transactivation. Based on the displayed results, it can indeed be suggested that luteolin activates Nrf2 to upregulated gene expressions of HO-1 & NQO1 even by the oral administration with extremely lower doses. Furthermore, it is interesting that these effects are clearly dependent on the timing of administration. Although the manuscript deserves further interest, some minor shortcomings and issues should be addressed.

MINOR

1/ Fig4: Why was NOT assessed the effects by lower dose administration like 0.1 mg/kg.

2/ Discussion section (P13line288): How were the values of 0.04-1.7 mg/kg calculated? I just wonder these should be 0.004-0.17 mg/kg, assuming human body wight is approximately 50 kg.

General

Please define all abbreviations at first use, eg BMAL1, CRY, & PER.

Title: Please improve it with correct English expressions.

Introduction:

P3line55: Please correct the spelling of Nrf2.

P3line65: …suggested that an Nrf2-dependent…

P4line78: …, our previous study…

6. PLOS authors have the option to publish the peer review history of their article (what does this mean?). If published, this will include your full peer review and any attached files.

Reviewer #1: No

Reviewer #2: No

---

## [Author Response · Author response to Decision Letter 0]

18 Mar 2020

Response to the reviewers

To the editor

Thank you very much for giving an opportunity to revise the manuscript. The comments from the reviewers are helpful to make a better manuscript. We revised the manuscript according to the comments. Please find our responses shown below. 

To the Reviewer 1

<Your comment>

This manuscript is very interesting for relationship of the luteolin-induced Nrf activation and circadian rhythms in vivo. In fact, author showed that low-dose luteolin activates Nrf2 and increase plasma luteolin levels in active phase. However, I have a few questions. Thus, I propose that to improve the manuscript and to increase potential impact figures in this field of scientific research.

<Our response>

We appreciate your reviewing our manuscript and giving your critical comments and useful suggestions to us. All of them are helpful for improvement of the manuscript. We addressed to all comments and suggestions and made the revised version of manuscript. Point-by-point responses are shown as follows:

<Your comment #1>

In Fig.1, author showed that protein levels of Nrf2 and HO-1 were induced in ZT12, but not ZT0. However, this can’t be investigated. Because Nrf2 levels in control level (luteolin 0 mg/kg) is different at ZT0 and ZT12. If author wish to compare the two groups, author should be measure the protein expressions on the same membrane.

<Our response>

Thank you for your comment. According to your comment, we carried out additional experiments to compare the difference of the basal expression of HO-1 and NQO-1. The data was replaced in Figure 1 and manuscript was revised (lines; 209, 216-217). Comparison of HO-1 between ZT0 and ZT12 was explained in Fig. 2.

<Your comment #2>

In Fig.2, author indicated mRNA level of Nrf2 was regulated by circadian rhymes. This point is very interested. However, I have a question. Why did Nrf2 protein level decrease in ZT12 compared to ZT0? Author should be measure Nrf2 levels in nuclear fraction and keap1 protein level in cytoplasm.

<Our response>

Thank you for your important comment. We carried out additional experiments to determine the expression of nuclear Nrf2 and cytosolic Keap1. As a result, nuclear Nrf2 and cytosolic Keap1 were higher at ZT0 than at ZT12 (Fig 2C, 2D). These results suggest that the transcriptional activity of Nrf2 was higher at ZT0 than ZT12 and Nrf2 activity is regulated by transcription of mRNA. However, the level of Nrf2 ubiquitination and interaction with Keap1 remain unclear. Further studies are needed in the future. This result was added in the result section and Figure 2, and the manuscript was revised (lines; 226-231 and 241-245).

<Your comment#3>

For statistical analysis, it is better to do a statistical test using one-way analysis but not Dunnett’s test. Because various concentration of luteolin treated for mice in this animal experiment. This point is very important.

<Our response>

According to your comment, the statistical analysis do again using one-way analysis of variance (ANOVA) with the Dunnett’s test as a post hoc test. The manuscript was revised (lines; 196-200).

To the Reviewer 2

<Your comment>

The authors of 'Low dose of luteolin activates Nrf2 in the liver of mice dependent on an administration timing: luteolin activates Nrf2 at the start of the active phase but not that of inactive phase of mice' present a structured manuscript, focusing on the effects of low dose of luteolin in mice liver, especially with regard to its regulation of Nrf2 transactivation. Based on the displayed results, it can indeed be suggested that luteolin activates Nrf2 to upregulated gene expressions of HO-1 & NQO1 even by the oral administration with extremely lower doses. Furthermore, it is interesting that these effects are clearly dependent on the timing of administration. Although the manuscript deserves further interest, some minor shortcomings and issues should be addressed.

<Our response>

Thank you very much for reviewing our manuscript. Your comments are helpful to make a better manuscript. According to your comments, we revised the manuscript. Please find following our responses as a Point-by-point response fashion.

<Your comment #1>

Fig4: Why was NOT assessed the effects by lower dose administration like 0.1 mg/kg.

<Our response>

Thank you for your comment. Since nuclear translocation of Nrf2 was the highest at 1 mg/kg luteolin, we evaluated the effect of 1 mg/kg luteolin on the mRNA expression of Nrf2 and its target proteins. The reason was added in result section (lines; 254-256).

<Your comment #2>

Discussion section (P13line288): How were the values of 0.04-1.7 mg/kg calculated? I just wonder these should be 0.004-0.17 mg/kg, assuming human body wight is approximately 50 kg.

<Our response>

Thank you for your important comment. The values of 0.04-1.7 mg/kg were calculated by using the human equivalent dose calculation formula (Nair AB and Jacob S, J Basic Clin Pharm. 2016; 7: 27-31.). We added the explanation in the text (lines 302-303) and a reference (Ref. #30). 

<Your general comment>

Please define all abbreviations at first use, eg BMAL1, CRY, & PER.

Title: Please improve it with correct English expressions.

Introduction:

P3line55: Please correct the spelling of Nrf2.

P3line65: …suggested that an Nrf2-dependent…

P4line78: …, our previous study…

<Our response>

According to your comments, the manuscript was revised (lines: 34-36, 56-57, 66 and 79). Title (line 1-2) was also changed.

---

## [Editor Report · Decision Letter 1]

24 Mar 2020

Low dose of luteolin activates Nrf2 in the liver of mice at start of the active phase but not that of the inactive phase

PONE-D-20-01855R1

Dear Dr. Ashida,

We are pleased to inform you that your manuscript has been judged scientifically suitable for publication and will be formally accepted for publication once it complies with all outstanding technical requirements.

With kind regards,

Nobuyuki Takahashi, Ph.D.

Academic Editor

PLOS ONE

Additional Editor Comments:

Authors properly addressed all of reviewers' comments.

So, I have decided to "accept" this manuscript without sending it to the reviewers again.

Reviewers' comments:

Nothing.

---

## [Editor Report · Acceptance letter]

27 Mar 2020

PONE-D-20-01855R1 

Low dose of luteolin activates Nrf2 in the liver of mice at start of the active phase but not that of the inactive phase 

Dear Dr. Ashida:

I am pleased to inform you that your manuscript has been deemed suitable for publication in PLOS ONE. Congratulations! Your manuscript is now with our production department. 

With kind regards,

on behalf of

Dr. Nobuyuki Takahashi 

Academic Editor

PLOS ONE